# Probiotics: If It Does Not Help It Does Not Do Any Harm. Really?

**DOI:** 10.3390/microorganisms7040104

**Published:** 2019-04-11

**Authors:** Aaron Lerner, Yehuda Shoenfeld, Torsten Matthias

**Affiliations:** 1B. Rappaport School of Medicine, Technion-Israel Institute of Technology, Haifa 3200003, Israel; 2AESKU.KIPP Institute, 55234 Wendelsheim, Germany; matthias@aesku.com; 3The Zabludowicz Center for Autoimmune Diseases, Sheba Medical Center, Tel Hashomer, Sackler Faculty of Medicine, Tel-Aviv University, Tel Aviv 5262000, Israel; shoenfel@post.tau.ac.il

**Keywords:** probiotics, autoimmune disease, horizontal gene transfer, virulent factor, antibiotic-resistant genes, public health

## Abstract

Probiotics per definition should have beneficial effects on human health, and their consumption has tremendously increased in the last decades. In parallel, the amount of published material and claims for their beneficial efficacy soared continuously. Recently, multiple systemic reviews, meta-analyses, and expert opinions expressed criticism on their claimed effects and safety. The present review describes the dark side of the probiotics, in terms of problematic research design, incomplete reporting, lack of transparency, and under-reported safety. Highlighted are the potential virulent factors and the mode of action in the intestinal lumen, risking the physiological microbiome equilibrium. Finally, regulatory topics are discussed to lighten the heterogeneous guidelines applied worldwide. The shift in the scientific world towards a better understanding of the human microbiome, before consumption of the probiotic cargo, is highly endorsed. It is hoped that better knowledge will extend the probiotic repertoire, re-confirm efficacy or safety, establish their efficacy and substantiate their beneficial effects.

## 1. Introduction

For a long time and mainly in the last decades the scientific, medical, industrial, and consumer communities have started to unravel the “superorganism” or “the second brain” presented by the prokaryotes dwelling inside the human enteric lumen [1,2,3]. The gut microbiome is a key player in intestinal eco-events, modulation, homeostasis, and function, dispersing systemically to impact human health [4,5]. Despite the bacterial part in the microbiome, it also contains viruses, archaea, fungi, and protozoa, reaching roughly 10^14^ cells, equaling the human body’s cell number and outnumbering the number of human genes by 100 fold. On the evolutionary aspect, microorganisms inhabited our globe 3.8 billion years ago, much earlier than the genus *Homo* emerged in Africa, 2.5 million years ago [6]. Thus, microbes had a much longer time outside us to adjust and adapt, by developing survival mechanisms enabling them to overcome even extreme environments. [7,8]. Notably, human enteric physiological microbiota composition and diversity, beyond infancy, are equilibrated, demonstrating amazing resilience to various perturbations, thus keeping it in the enteric compartment for the benefit of both kingdoms [9].

The situation is completely different with the much later evolutionary introduction of probiotics, most probably during the Neolithic layer of the stone age period, nearly 10,000 years ago [10,11]. Louis Pasteur and E. Metchnikoff should be acknowledged for further defining probiotics toward their actual definition “live microorganisms, which when administered in adequate amounts, confer a health benefit on the host.” Since then numerous publications expanded on the beneficial aspects of probiotic intake on human disease, conditions, mood, behaviors, and performances [12,13,14,15,16]. Just to cite a recent example: “The strongest evidence in favor of probiotics lies in the prevention or treatment of five disorders: necrotizing enterocolitis, acute infectious diarrhea, acute respiratory tract infections, antibiotic-associated diarrhea, and infant colic” [15]. The repertoire of the most consumed probiotics contains lactic acid producing bacteria, mainly *lactobacilli, Bifidobacterium, lactococci,* and *streptococci*. Yeast, bacilli, and some non-pathogenic *E. coli* strains are less often used. The ingested and nutritional, industrially used probiotics and starter cultures were generally considered as safe, for many years, but recently a change in trend has appeared.

The present review will summarize the questionable or unsubstantiated benefits of probiotics in a meta-analysis, highlighting their negative side, side effects, and expanding on their potential harmful capacities, mechanistic pathways, and potential hazards for human health.

### 1.1. The Probiotic Market is Booming Sky High

According to several reports, probiotics were used, unintentionally, nearly 10,000 years ago, but spread as fermented milk in previous centuries, mainly in Europe, where in the Balkan regions life span longevity and healthy life were attributed to their fermented yogurt. Their food application expanded earlier and on a larger scale compared to the sales in sachets, capsules, or other pharmaceutical preparations [17]. It seems that the trend is changing since direct personal consumption is surging abruptly alongside the adoption of self-care, integrative medicine, social enterprises, and aggressive media advertisement [18,19]. In fact, according to global market analysts, the global probiotic market size is predicted to exceed 3 billion US dollars by 2024 [20]. The market size for *lactobacilli* strains was valued at 1.2 billion US dollars in 2017, while the *Bifidobacterium* market size prediction will increase close to 6% until 2024 and the *Bacillus* strain market size may surpass 180 million US dollars by 2024 [21]. According to the discussion group at the 2017 meeting of the International Scientific Association for Probiotics and Prebiotics, the estimate of product creation and sales will reach 50 billion US dollars within the next five years [19].

### 1.2. Probiotic: Medical and Non-Medical Uses

The medical and non-medical indications to consume, over the counter, probiotics is constantly increasing. Table 1 summarizes some of those applications.

The present review is not intended to cover or update on the various indications for probiotic ingestion but will highlight their usage as a supplement to pharmaceutical therapy of, for example, autoimmune conditions.

### 1.3. Can Probiotic Intake Change Microbiotic Composition and Restore Eubiosis?

The holistic enteric compartment with its active luminal organ, presented with abundant prokaryotic flora, creates multifunctional defense mechanisms, preventing pathogenic invaders while keeping the microbiome at bay [5,8,9]. However, when failed or transformed to the dysbiome, chronic pathological conditions evolve, resulting in allergic, inflammatory, cancerous, and autoimmune diseases. The hen and chicken dilemma remains an enigma, and the issue of association/causality is far from being resolved. Despite this, taking probiotics to restore the normal, protective microbiome and to “balance” the human body’s flora is booming. The question arises over whether probiotics can modify dysbiosis and reverse the process. According to many recent scientific observations, the ways that the enteric microbiota/dysbiota is influenced by “good bacteria,” or the probiotic, is still unknown, and it seems that probiotic intake does not significantly affect gut flora composition [39,40,41,42,43,44,45,46,47,48,49]. Interestingly, probiotics affect the upper small bowel more than the colon [41,45]. Two major aspects ensue from recent studies: 1. The effects are influenced by the individual microbiome composition and structure, eluding to personal medicine [40,42,43,45]; 2. The probiotics impact is much more related to functional aspects, praising the mobilome, bacterial constituents, metabolomics, as well as the proteinomic effects [40,41,42,43,44,45,46]. After setting the stage of the probiotic clinical usages and impacts, the following is a clinical example of probiotics and the autoimmunity relationship.

### 1.4. Probiotics as a Supplemental Therapy in Autoimmune Diseases

The etiology, pathomechanisms, initiation, maintenance, progression, and modulation of autoimmunogenesis are continuously evolving. On the other hand, it is not yet understood why 5–8% of the general public is affected by autoimmune conditions. Four mechanisms were suggested for autoimmunity generation: molecular mimicry, self-antigen modification, bystander activation, and immune reactivity modulation [49]. The place of the wide “exposome” and more specifically the dysbiosis associated autoimmune disorders, is far from being clarified. Noteworthy, the relations of the changed composition and diversity in the enteric microbial kingdom to the four pathogenic mechanisms enumerated above is still poorly understood. Despite the lack of the dysbiotic scientific background, the patients and the treating clinical teams are trying to advance the supplemental therapeutic modalities in the direction of restoring the altered microbiome. The application of fecal transplantation, prebiotics, and probiotics are constantly expanding. Multiple autoimmune conditions are treated by probiotics including systemic lupus erythematosus (SLE), rheumatoid arthritis, Celiac and Crohn’s diseases, ulcerative colitis, multiple sclerosis, Sjogren’s syndrome, systemic sclerosis, antiphospholipid syndrome, myasthenia gravis, diabetes type 1 [13,50,51,52,53,54,55,56,57,58,59,60,61,62]. In many local and international scientific conferences, the probiotics trial as a supplement of adjunct therapy is presented. Moreover, the main argument favoring it is: “If it does not help it does not do any harm.”

The purpose of the current review is to protest against that repetitive declaration and to justify the opposite of its dual messages: 1. According to multiple recent meta-analyses, probiotic clinical benefits are questionable or disqualified. 2. Probiotics can present a Trojan horse that works against human health.

## 2. The Dark Side of Probiotics

Microbes possess an extended arsenal of hostile factors, capable of suppressing or destroying vital eukaryotic host mechanisms, for their advantage. Probiotics are an integral part of the prokaryotic kingdom with evolutionary conserved self-survival systems, in ex vivo and in vivo environments; above all, mainly in the overpopulated, extremely competitive, harsh ecological niche of the human gut. Ingestion of probiotic bacteria or products creates a survival struggle between the well-established inhabitant microbiome. Microbiome in the enteric compartment with the new probiotics.

In addition to the clinically reported multiple side effects in the literature, several pathogenic virulent potential pathways can be expressed in and executed by probiotics, thus affecting human health integrity. The following details several such pathomechanistic avenues.

### 2.1. Horizontal Gene Transfer (HGT)

HGT is the lateral movement of mobile genetic elements between unicellular or multicellular organisms. It enables the transfer of genes even between distant species mediated usually by transformation, transduction, conjugal transfer, or with specific gene transfer agents [63]. The topic of HGT in the human gut and the transfer of virulent genes to the endogenous microbiome was summarized recently [8]. The human gastrointestinal tract is an ideal environment and represents a hot spot for HGT [8]. As probiotics are extensively used in the processed food and fermented product’s industries and as over the counter additives, the question arises whether they can deliver hostile genetic elements to the microbiome?

Screening the literature, multiple publications describe the existence and transfer of hostile mobile genetic elements in and from probiotics [8]. Taking, for example, the most explored ones, the antibiotic-resistant genes, were found in various dietary supplements [64]. The problem is so widespread that it requires risk assessment measures to be implemented in those nutritional supplements [65]. More so, virulent mobile genetic elements are of a concern when transferred by HGT from probiotics to the enteric commensal communities [66]. More specifically, HGT between probiotic strains was reported for *Lactobacillus paracasei* [67], *Lactobacillus rhamnosus* [68], *Lactobacillus reuteri* [69,70,71], *Lactobacillus gasseri* [72], *Lactobacillus plantarum* [71], among other probiotics. Generally, gene flux of antibiotic-resistant genes, from gram-positive cocci to gram-negative microbes has been suggested [73], involving numerous antibiotics [74]. Even if probiotic ingestion does not impact stool microbial composition [48], HGT between ingested probiotic and the endogenous dwellers exist [75]. The cumulative risk of the probiotic double-edged sword effect of lateral genetic transfer of virulent elements is an ongoing enigma [76].

A special compliment should be given to Rosander et al. [77] who wrote a rare publication on the removal of antibiotic resistance gene-carrying plasmids from *Lactobacillus reuteri* ATCC 55730, which is not commonly reported in probiotic research. However, antibiotic gene transfer is only one aspect of virulent genes and was taken just as an example. Gelatinase and hemolytic activities and several enzymes like peptidases, acid phosphatase, phosphohydrolases, α + β- galactosidases, and *N*-acetyl-β-glucosaminidase were depicted in lactic acid bacteria of aquatic origin intended for use as probiotics in aquaculture [78]. Most recently, microbial transglutaminase, a heavily consumed additive by the industrial processed food industry and a prokaryotic survival factor, was recently found to possess virulent factors, with anti-phagocytic being one of them [79,80]. Interestingly, probiotics also secrete the enzyme that was described as a novel potential environmental factor in celiac disease induction [81,82,83,84,85]. Microbial transglutaminase can be considered as a secreted toxin [86], with functional capacities even in pathogenic microbes [87,88,89,90]. Complexed to gliadin, when the microbial transglutaminase is crosslinking gliadin, the complex is immunogenic in celiac patients [84], and multiple deleterious effects on human health were described [91]. Finally, lateral gene transfer might influence the holobiont repertoire in intestinal niches whereby external prokaryotes, including probiotics, can affect genetic stability and evolutionarily conserved processes, threatening human health [92,93,94,95,96,97,98,99].

### 2.2. Bacteriophages of Probiotics Transfer Mobile Virulent Genes

The success and efficiency of probiotics depends on numerous factors that can be divided into microbiotic-exogenous, host-endogenous, and luminal-environmental [100]. One of the luminal factors are the bacteriophages. They are bacterially infectious small viruses that lyse microbes. The gastrointestinal tract harbors a wide variety of viruses, called the virome, and the phageome constitutes the largest part of this virome [101]. It is estimated that more than 30 billion bacteriophages transcytose human epithelial layers every day. They play a pivotal role shaping the microbiome’s taxonomic and functional compositions. The enteric prophages serve as a mobile repository of genetic elements and are transmitted via our microbiome, thus impacting on the microbiota/dysbiota or symbionts/pathbionts ratios and health and disease [101,102]. The enteric phageome virulence is controlled by the neighboring microbes, fungi, and helminths, thus creating a luminal trans-kingdom relationship [103,104].

They can be regarded as human pathogens, interacting directly or indirectly with prokaryotic, probiotics, as well as eukaryotic cells, including involved in protein misfolding, carrying prion-like domains [102,105]. Probiotics are prokaryotic and as such, are under the influence of the bacteriophages [100,101,102,103,104,105], many of the probiotic strains, orally consumed or used in food industries have their specific phages. A broader view will disclose a global environmental distribution of bacteriophages, carrying their hostile genetic cargo, to most environmental biomes where the bacteria reside [106,107]. Wastewater treatment plants, human fecal samples, food and medical isolates, dairy fermentations, agriculture, and even in the air are where virulent genes were most recently detected in multiple congested metropolitan urban air [108,109,110,111,112,113]. Screening the bacteriophages of commonly used probiotics, *Lactobacillus paracasei* or *gasseri* [67,72,114,115], *Lactococcus lactic* [116,117,118,119,120,121], and many more, were found to have close contact with specific phages.

The probiotic bacteriophages are a potential carrier of hostile genes that by transfecting prokaryote or eukaryote cells, can spread genetic material. Finally, it appears that some bacteriophages contain virulent transglutaminase genes, thus representing additional tranglutaminase activity in the intestinal lumen [86,90], in addition to the microbiome, archaeal, probiotic, and industrially added one [8,82,83,84,85,91]. Finally, the evolution of novel transglutaminase-like peptidase from eukaryotic ciliary compartments was traced back to prokaryotic transglutaminase-like peptidases, thus, deciphering key evolutionary events along the course of the eukaryotic emergence from prokaryotes [122].

### 2.3. Processed Food and the Probiotic Mobilome

Probiotics are heavily used in the processed food industries, spanning not only dairy fermented products, but also in wider industrial applications including medical, diagnostic, pharmacological, and biotechnological industries [123,124,125,126,127,128]. For many industrial applications, including dairy starter fermentation cultures, they acquired the GRAS (generally regarded as safe) status, which was defined before recent safety concerns were raised, such as the carriage of virulent mobile genetic elements. Notably, genetic transfers in bacteria are more prone to occur in crowded environments, such as the human GI tract, not excluding food reservoirs, manipulations, and products. Probiotics are heavily used, for many years, for processed food manipulations and production. Many of them are used in fermented foods like dairy products, cheese, fermented sausage, fermented vegetables, soy-fermented foods, and fermented cereal products [129]. It is very logical that one of the concerns of their massive usage is the lateral exchange of hostile genes, in-between them, or to the physiological microbiome, to the dysbiota, or even to human cells, as summarized here [8]. Gene acquisition/loss within or between various microbes and probiotic strains were widely described, all across the food chain, be it dairy, meat, or vegetable products and even in the ready-to-go food items [121,130,131,132,133,134]. HGT of antibiotic resistance is wide and was reported for a wide range of probiotics, including *Lactobacillus rhamnosus*, *Lactobacillus gasseri*, *Lactobacillus paracasei*, *Lactobacillus reuteri*, *Lactobacillus plantarum,* and many others [8]. In addition to antibiotic-resistant genes, the most extensively explored, numerous additional virulent genes are carried by the probiotic microbial genome. Microbial transglutaminase, mentioned above, is only one of them [79,80,81,82,83,84,85,86,87,88,89,90,91].

### 2.4. d-lactate, Metabolic Acidosis, and Brain Fogginess

Probiotic consumption is associated with D-lactic academia and acidosis in adults and infants on probiotic-containing formula [41,135,136,137,138,139]. *Lactobacillus* and *Bifidobacterium* species are the most used bacteria in probiotic formulations and they produce d-lactate [139,140,141,142] and their consumption was suggested to be avoided in d-lactic acidosis [139]. Intriguingly, d-lactic acidosis and other etiologies for acidosis are associated with neurocognitive symptoms, neurological impairments, and chronic fatigue syndrome [137,143,144], including brain fogginess [41,139]. The syndrome of brain fogginess has, in fact, multiple etiologies, one of which is short bowel syndrome associated with D-lactic acidosis [41,139,144,145]. Despite the critical view on the association with probiotic intake [146,147], the discontinuation of the antibiotics and the resolution of the symptoms on antibiotic therapy, strengthen the causative association [41,139].

### 2.5. Intestinal Bacterial Overgrowth, Gas, and Bloating

This paragraph is related to the above paragraph and might explain the pathophysiology of the acidosis and brain fogginess described above. Rao et al. described a new syndrome relating post-prandial brain fogginess, gas, and abdominal bloating to small intestinal bacterial overgrowth and probiotic-induced d-lactic acidosis [41,139]. Additional complaints were fatigue, weakness, disorientation, and restlessness. The authors put forward the hypothesis that probiotic fermented carbohydrates in the proximal small bowel induce intestinal bacterial overgrowth, resulting in d-lactic acid production, increased gas output, and abdominal bloating. The d-lactic acidosis is the culprit for the brain fogginess. They suggested that this unique entity is an additional side effect of probiotic consumption.

### 2.6. Additional Clinical Probiotic Side Effects

The medical literature warns against probiotic consumption in congenital or acquired immune debilitating conditions, heart anomalies, chemo- and radiotherapies, surgical abdomen, HIV-infected, critically ill, post-organ transplantation, post-operation, central venous catheters, autoimmune disease on immune suppression, pregnancy, neutropenia, critically ill patients, including antibiotic-associated diarrhea, active ulcerative colitis, and potential for translocation of probiotic across bowel wall [74,148,149]. Although it is not the main focus of the present review, to wrap up the subject, Table 2 summarizes the reported toxic, unintended, adverse effects following probiotic usage.

The list of probiotic’s adverse effects is expanding, however, due to a lack of safety and toxicity standardized protocols and regulatory implementations, the list is under-representative. It is clear that more safety and toxicity designed studies are needed to reveal the negative side of probiotic use [74,148,149,154,165]. Figure 1 is a schematic presentation of the local and systemic adverse effects and mechanisms by which the probiotics exert their deleterious effects.

## 3. Problematic Inadequate Design, Incomplete Reporting, and Lack of Transparency

The current review aims to highlight the negative side of probiotic consumption. As such, followed herein are the most recent systemic reviews and meta-analyses that criticize multiple aspects of the medical publications on probiotic efficacy and safety (Table 3). Many of them detected a lack of qualified experimental designs, a shortage of standardization, extended data variance, incomplete reporting, high patient withdrawal, and all of which increase the publications’ biases. In a recent review of existing meta-analyses, the authors tried to analyze the contradictory results of the probiotic effectiveness in many frequent conditions [165]. The final results were quite restricted: “Only for antibiotic- and *Clostridium difficile*-associated diarrhea, and respiratory tract infections the effects of probiotics are considered "evidence-based". Concerning other fields, meta-analyses fail to define the type and biologic effect of probiotic strains, as well as the outcome in a disease state. The authors concluded that: “Further studies are needed, because the available evidence is insufficient to show the efficacy of probiotics themselves. Carefully designed clinical trials are needed to validate the effects of particular strains of probiotics given at specific dosages and for specific treatment durations.”

## 4. Lack of Effective Regulation of Probiotics

More recent systemic reviews or meta-analyses, from 2018, did not demonstrate differences of outcomes, using probiotics, on the treatment success of: constipation [180], traveler’s diarrhea [179], cancer [173,175], anxiety [171,177], rheumatoid arthritis [62], urinary tract infections [174], decrease in fat mass [172], food allergy [169], childhood asthma [178] and eczema [185], preterm neurodevelopment [182], and adiponectin and leptin levels [181]. Van den Nieuwboer et al. summarized it clearly: “generalizability of conclusions are greatly limited by the inconsistent, imprecise, and potentially incomplete reporting as well as the variation in probiotic strains, dosages, administration regimes, study populations, and reported outcomes” [148].

It seems that scientific and medical societies should “mind the gaps” between published studies praising probiotic therapeutical efficacy and a lack of substantiation when analyzed by more objective, standardized methodologies, such as critical systemic reviews or meta-analyses. More so, when intestinal microbiota composition was assessed on probiotic intake, no significant changes were depicted [44,45,46,47,48,140]. Intriguingly, probiotic effects diminish with time, in mice [186] and their stability and survival markedly decline in frozen capsules [187]. Probiotic colonization is also controversial since studies done on fecal samples alone are insufficient—colonized intestinal biopsies are more indicative [149].

## 5. Probiotic Safety is Under-Reported

The present review cannot be completed without an update on probiotic safety. The food and drug administration consider some probiotics, as GRAS, when added to food [188], especially when intended to impact taste, aroma, or nutritional value [74,189]. However, most of the reviews, analyzing the safety of probiotics highlight the issue of a lack of structural classification and a wide generalization of conclusions that is limited by imprecise, inconsistent, and incomplete reporting intermingled with variations in strains used, dosages, regimes of administration, experimental designs, and the participating populations [74,148,149,167]. Many contrast the differences that exist between the long history of the large consumption of “safe” probiotics compared to the scarcity of scientific proof for their safety [74,149,167]. Missed microbial identification, misnumbering and mislabeling, and lack of dose-response relationships are additionally reported aspects [149,190,191,192,193,194]. Insufficient and uncontrolled research designs, underpowered studies, and mixed research and outcome results are often encountered [195]. The allergic reaction, or anaphylaxis, is an additional aspect of safety, since probiotic preparations can contain allergens, including cow milk and hen egg proteins [196,197]. Safety is further complicated by the fact that various companies use duplicate cultures of the original strains, by applying fingerprinting techniques, potentially increasing the risk of detrimental effects [149]. To fill the gap in reliability and transparency in probiotics effectiveness and safety the following need to be considered: eliciting side effects data from participants [198], suboptimal adherence to reporting guidelines [199,200,201], over food industry funding mounting to 60% of the screened studies [202], lack of long-term effects in normal and vulnerable populations [203], and occasional lack of viable organisms [204]. Most recently, skepticism was raised concerning the labeled number of bacteria in probiotic preparations, publication bias, the generalizability of findings, and the safety in immune deficient hosts [13]. Finally, it appears that few studies on probiotics are designed to probe safety aspects and much should be improved in this domain [149,205,206].

Quite often patients consume probiotics while physicians encourage probiotic consumption, despite their potentially harmful effects. The notion of “good bacteria” and the manufacturers’ claim of “health promotion” or “balancing” normal gut flora should be taken with a “grain of salt.” In two seminal studies from an Israeli group recently published [42,43] it was shown that “mucosal colonization resistance to empiric probiotics” is host and microbiome dependent and that “post-antibiotic gut mucosal microbiome reconstruction is impaired by probiotics” [40]. Those and multiple reviews, meta-analyses, and studies, mentioned above, reinforce the need for regulation of probiotics for public health protection. It seems that real life habits, hopes, and media-directed information overcome scientific knowledge in real-time. Even the basic categorization of probiotics as drug, food, or dietary supplements is still undetermined and confused [30]. However, there is some light in the regulatory tunnel. The European Food Safety Authority (EFSA) changed their regulatory policy based on the lack of convincing evidence on the claim that probiotics improve human health or wellbeing [195,207,208]. Moreover, when regulation is enforced, consequences are predicted. Within the European Union, all health claims for probiotics were rejected, except for lactose intolerance improvement [209]. Since 2013, no claims concerning the change or improved gut microbiome composition was approved by the EFSA. The American FDA is taking a different approach. Probiotics can be categorized as food, food additive, cosmetics, dietary supplement, or drugs [210], and the responsibility for accuracy and truthfulness of the product is the responsibility of the producer [209,211]. It should be notified that no probiotic was approved for health claims by the FDA in recent years [209]. The tightened regulations impacted the scientific community and the manufacturer’s policies profoundly, as mirrored by the number of publications on the subject in the last two decades. The number of publications or registered studies increased significantly on microbiota while plateauing on probiotics [209]. The shift to explore the microbiota presents an opportunity to uncover new probiotics and understand their mode of action, and explore relationship with their neighboring prokaryotes, eukaryotes, and their secreted mobilomes [212,213].

## 6. Conclusions

The present review intended to summarize the somber side of probiotics, highlighting the potential detrimental effects embedded in the fact that probiotics are prokaryotes, and as such, contains hostile factors, in order to survive. They are capable of inducing local and systemic adverse effects (Figure 1) thus contradicting their definition as beneficial for human health. More caution, safety exploration, and stringent regulation can prevent these mal-effects. The absence of associated virulence factors should be demonstrated, especially when the probiotic belongs to a bacterial genus with pathogenic capabilities. Consideration of risk-benefit ratio before suggesting probiotics should be highly recommended. In view of potential pathogenic pathways, problematic inadequacy of design, reporting and transparency, and under-reported probiotic safety and non-defined implementable international criteria for regulation, it is encouraging to follow the contemporary back shift to the microbiome. It is hoped that by widening the knowledge of the human intestinal microbiome, that salvation will come from “the ascent of the blessed” probiotics as a preventive/beneficial/therapeutical health promoter. Finally, since intestinal microbiota is a recent new frontier in medicine, further exploration might stage probiotics as a preventive barrier or as a product capable of balancing the dysbiome associated with chronic human morbidity and mortality.

## Figures and Tables

**Figure 1 microorganisms-07-00104-f001:**
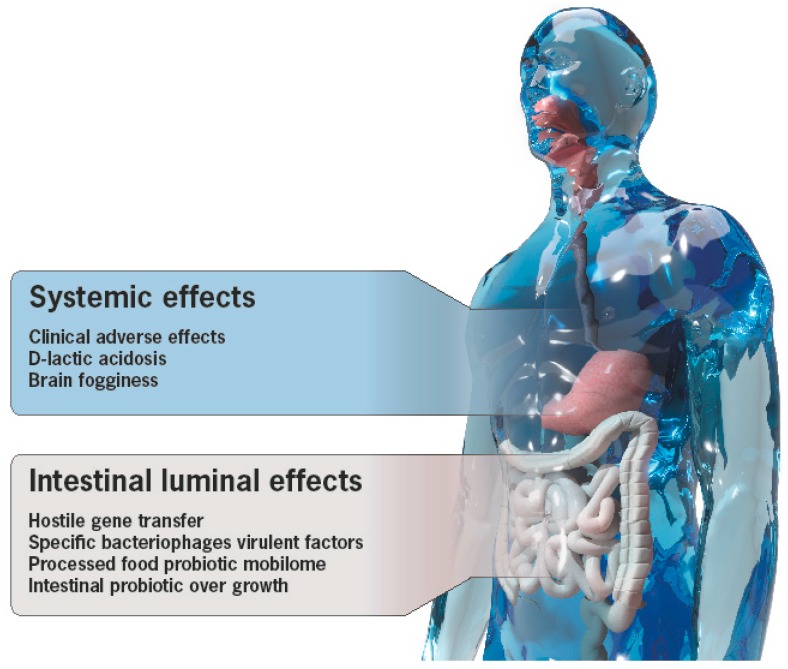
The local and systemic adverse effects of probiotics.

**Table 1 microorganisms-07-00104-t001:** Medical and non-medical indications for probiotic consumption.

Medical Applications 1	Medical Applications 2	Functional Applications
Lactose intolerance [20,22,23]	Chronic renal failure [22,24]	Functional digestive complains [22]
Hyperlipidemia [18,22]	HIV infection [25]	Mood and behavior changes [26]
Nephrolithiasis (oxalate stones) [27]	Cirrhosis, liver encephalopathy, NAFLD [15,23]	Memory improvement [28]
Inflammatory bowel disease [22,23,24]	Organ transplant [23]	Anxiety, fatigue, weakness, body or localized pains, nausea [26,29]
Irritable bowel syndrome [13,22,23,30]	Metabolic diseases [12,22]	Constipation/loose stools changes [22]
Eczema, allergic rhinitis, asthma [12,22,24,30]	Constipation [22]	Day care health [24]
Infectious diarrhea [22,23,24,30]	Periodontitis [22]	Working places health [24]
Respiratory tract infections [12,22,24]	Depression [26]	Wellbeing [17,24]
Traveler’s diarrhea [22,23,24]	Stay in Intensive care unit [31]	Anti-stress [29]
Necrotizing enterocolitis [13,24]	Prematurity [32]	Increase longevity [33]
Pouchitis [34]	Infant colic [13,15]	Improve sexuality [35]
Helicobacter pylori [22,30]	Autoimmune diseases [13,22,23,24,30]	Impaired “intestinal integrity” [22,24]
Neurological disorders [21]	Cystic fibrosis, pancreatitis [23,30]	
Overweight and obesity [18,21]	Ethanol-induced liver disease [23]	
Various cancers [22,23,30]	Small bowel bacterial overgrowth [22]	
Along or after antibiotics therapy [22]	Enhancement of oral vaccine administration [30]	
*Clostridium difficile* induced colitis [22,23,30]	Ischemic heart disease [18,22]	
Respiratory/urinary tract, rotavirus infections [22,23,24]	Hypertension [36]	
Vaginosis [24,30]	Neuropsychiatric/degenerative diseases [37,38]	
Dental caries [22,23,24,30]	Enhance growth [22,24,39]	
Diabetes type 2 [23]	Enhance weight gain [22,24,39]	

**Table 2 microorganisms-07-00104-t002:** Summary of the reported toxic, unintended, adverse effects following probiotic consumption.

Infectious/Gastrointestinal	Allergic	Genetic	Patho-Toxogenicity
Bacteremia [41,74,150]	Rhinitis [149]	Transfer of virulent factors:	Enhanced adhesion and protein aggregation [74,149]
Sepsis [41,74]	Wheezing bronchitis [151]	Antibiotic resistance [74,149,152,153,154]	Mucolysis/hemolysis [74,149]
Fungemia [41,155]	Rash [149]	Hemolysin [149,152]	Bile salt hydrolysis [74]
Endocarditis, meningitis, endometritis, peritonitis, pneumonia [150,156,157]		Gelatinase [149]	DNA degradation and proteolysis [149]
Liver abscess [150]	Metabolic	DNAse [149]	Innate defense resistance [52,149]
Diarrhea, Abdominal cramps [74]	d-lactic acidosis [41,74,149]	Enolase activating plasminogen [149]	Food poisoning [149]
Nausea, vomiting, flatulence, taste disturbance [41,74]		Metalloendopeptidase [158]	Immune evasion or over stimulation [74,149]
Low appetite [159]		Cytolysin modification, transport, activation [160]	Facilitated microbial conjugation/translocation [74,149]
		Sex pheromones [161]	Macrophage/monocyte chemotactism [162]
			Nanoparticles: Lactomicroselenium [163]
			Gastrointestinal ischemia [41,74]
			Mechanical choking [74]
			Peptide deamidation [164]
			Epigenetic and mobilome manipulation [52]

**Table 3 microorganisms-07-00104-t003:** A summary of recent meta-analyses and systemic reviews criticizing microbiome and probiotic publications.

Publication	Mal-Designed	Lack of Standardization	High Data Variance	Biased	High Withdrawal	Incomplete Reporting	Reference
Review	+	+	+	+	+		[154]
Systemic review	+	+	+		+	+	[148]
Review	+	+	+	+		+	[74]
Systemic review	+	+		+		+	[166]
Systemic review	+	+	+	+		+	[148]
Systemic review	+	+	+	+		+	[167]
Meta-analysis				+	+		[168]
Meta-analysis	+			+		+	[169]
Meta-analysis	+		+	+	+		[170]
Meta-analysis	+			+			[171]
Meta-analysis	+	+	+	+			[172]
Meta-analysis	+	+	+	+			[173]
Systemic review	+			+			[174]
Meta-analysis	+	+		+		+	[175]
Meta-analysis	+	+		+			[62]
Meta-analysis	+	+		+			[176]
Meta-analysis	+	+					[177]
Meta-analysis	+	+		+			[178]
Meta-analysis	+	+		+			[179]
Meta-analysis	+			+			[180]
Meta-analysis	+	+	+	+			[181]
Meta-analysis	+	+		+			[182]
Systemic review	+	+	+	+			[183]
Meta-analysis	+	+	+	+			[184]

+ = exist in the publication.

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
