# Peer review of "Probiotics: If It Does Not Help It Does Not Do Any Harm. Really?"

_microorganisms, 2019, doi:10.3390/microorganisms7040104_

Round 1

Reviewer 1 Report

Excellent work. I believe that this review will offer significant outcome to the research communtity since more and more concerns are arrising lately regarding teh proper selection and applications of various candidate probiotics in food fermentations (dairy, vegetables etc). I strongly recommend "accept". 

Author Response

Thank you for your review and for your complements. Corrections were made according to reviewers 2 and 3. All the changes were highlighted in yellow. 18 references were added.

Reviewer 2 Report

1. Table 3. The authors present some publications to be against probiotic publications but this is not true. For example Ref 151 do not criticize probiotic publications at all. The only similar to what the authors state is “Although severe systemic side effects post-microbiome therapy are not observed to date, minor gastrointestinal discomforts such as abdominal pain, vomiting, nausea are commonly reported. This subsequently leads to withdrawal or non-compliance by patients during field trials. Therefore, improvement on treatment regime, route of administration, and effective communication with the patients are strongly encouraged.” But in my opinion this is not a criticism to probiotic publication, the opposite. In this work (ref 151) the authors believe, as the title says, “The Human Gut Microbiome – A Potential Controller of Wellness and Disease”. I cannot check the whole table. Please check all the references in this table in order to express what the authors really say.

2. Where is the figure 1? Without that figure the review process is incomplete.

3. Conclusion. This section is usually without references.

4. Table 1. The ref should not be at the title but after each indication in the table.

5. Table 2. The ref should not be at the title but after each effect in the table. This will help readers to know what each ref state.

Author Response

Thanks for your valuable comments, corrections and suggestions.

In the attached revised manuscript, all the corrections and added text and references are highlighted in yellow. 18 references were added. English was improved.

References were added into table 1+2 

references were deleted from the conclusions

I apologize for figure 1, the editorial office forgot to add it

Concerning table 3:

*In reference 151 (now reference No 164) the authors described "research gaps and limitations" in the gut microbiome research: "lack of direct evidence and mechanistic details", "our understanding on gut microbiome is still at a very preliminary stage, whereby there are several limitations and research gaps...", Hence, experimental design should be revised and tailored....", They question if the microbiome play a primary or secondary roles? "suspecting the existence of confounding factors, for example, subject-specific differences or distinctive experimental designs across studies becomes important", many subject-specific factors..... serve a crucial role in shaping the unique composition of human microbiota in each individual", "....make it much more challenging to design constructive and conclusive experiments to test the role of the microbiome....", "....the experimental design, differences across studies, standardization of research protocols are required to enable effective comparison of finding across studies, reduced data variance and biased......"

It is correct that the author suggest way to improve the gaps and limitations in the enteric microbiome studies, but indirectly, they criticize the published literature and mention : mal-designed, unstandardized, over variance, bias and increased subject withdrawal in the microbiome studies. The conclusions are optimistic, but, there is a substantial hidden criticism.  Those are the raison why reference 164 was introduced in table 3.

I appreciate your review and your contribution to the quality of the manuscript.

Thanks.

Reviewer 3 Report

Generally, I very much enjoyed this paper. It was indeed an excellent overview balancing the "generally accepted" notion that probiotics are always "good" or at least no risk. It will be challenged but such discussion is always good in science.  It is a short paper but has an extensive reference listing that should prove very valuable as a review. One aspect, however, that was left out, was a discussion of synbiotics. Does this mean that the benefits recorded for such products are always due to the prebiotic, not the probiotic component? Perhaps this is a question for future discussion.

I found the paper rather difficult to read and follow, perhaps due to the excessive use of medical jargon and terminology when more general scientific terms could be used to make it appeal to a wider audience, but perhaps that was my failure, not the authors'.

Minor sentence structure and wording corrections are needed as described below:   

Line 13: writing should be past tense: replace “…consumption is tremendously…” with  “… was tremendously”

Line 22:  use of the word “blessed” is colloquial, try “endorsed”

Line 29: writing should be past tense: replace “…communities start to unravel…” with  “… have started to unravel…”

Line 31: use of the word “irradiating”, please use another term

Line 32: Use of the word “microbial” should be “bacterial”, also, what about fungal

Line 34:  replace “…100 folds.” By “…100 fold.”

Line 36: “bugs” is colloquial. Either define or replace with “microbes”

Line 39:  use of the term “…in bay…” is colloquial, please reword

Lines 67-68: “ …for Bifidobacterium market size gains prediction, close to 6%.....” unclear, please reword

Lines 73-74:  Awkward sentence structure, please reword

Line 153:  the word “gens”  please revise

Line 155: check spacing/hyphenation 

Line 168:  change “…divided to…”  try “….divided into….”

Line 199:  “…only the dairy…”  revise to “…..only dairy….

Line 200: “….including in medical…”  revise to “…including medical…”

Line 207:  “…of them in fermented…”  revise to “…of them are used in fermented…”

Line 338: “…capable to induce local….”  Revise to “…capable of inducing local…”

Line 347: “…microbiome, the salvation…” revise to “…microbiome, that salvation…”

Line 349-350:  “….probiotics as preventive barrier or capable to balance…” revise to “…probiotics as a preventive barrier or capable of balancing ….”

Line 352:  The cited Figure No.1 was not attached to the manuscript.

Author Response

Thanks for your valuable suggestions, corrections, comments and complements.

All your suggestions were imbedded in the text and highlighted in yellow.

The review is concentrating on probiotics side effects and is not intended to discuss prebiotics and symbiotic and their mode of action 

We did not used too many medical "jargons" and we tried to deliver a friendly scientific languish

Thanks for improving the quality of the manuscript.

Round 2

Reviewer 2 Report

I think that is Ok in the present form and may be accepted.